# An Untargeted Metabolomic Analysis of *Lacticaseibacillus* (*L*.) *rhamnosus*, *Lactobacillus* (*L*.) *acidophilus*, *Lactiplantibacillus* (*L*.) *plantarum* and *Limosilactobacillus* (*L*.) *reuteri* Reveals an Upregulated Production of Inosine from *L. rhamnosus*

**DOI:** 10.3390/microorganisms12040662

**Published:** 2024-03-26

**Authors:** Luca Spaggiari, Natalia Pedretti, Francesco Ricchi, Diego Pinetti, Giuseppina Campisciano, Francesco De Seta, Manola Comar, Samyr Kenno, Andrea Ardizzoni, Eva Pericolini

**Affiliations:** 1Clinical and Experimental Medicine PhD Program, University of Modena and Reggio Emilia, 41125 Modena, Italy; luca.spaggiari@unimore.it (L.S.); francesco.ricchi@unimore.it (F.R.); 2Department of Surgical, Medical, Dental and Morphological Sciences with Interest in Transplant, Oncological and Regenerative Medicine, University of Modena and Reggio Emilia, 41124 Modena, Italy; natalia.pedretti@unimore.it (N.P.); samyr.kenno@unimore.it (S.K.); andrea.ardizzoni@unimore.it (A.A.); 3Centro Interdipartimentale Grandi Strumenti, University of Modena and Reggio Emilia, 41125 Modena, Italy; diego.pinetti@unimore.it; 4Institute for Maternal and Child Health-IRCCS, Burlo Garofolo, 34137 Trieste, Italy; giuseppina.campisciano@burlo.trieste.it (G.C.); mcomar@units.it (M.C.); 5Department of Obstetrics and Gynecology, IRCCS San Raffaele Scientific Institute, University Vita and Salute, 20132 Milan, Italy; fradeseta@gmail.com; 6Department of Medical Sciences, University of Trieste, 34129 Trieste, Italy

**Keywords:** *L. rhamnosus* (*L.* RHA) metabolome, *L. acidophilus* (*L.* AC) metabolome, inosine, untargeted metabolomics, postbiotics, cell-free supernatants (CFS)

## Abstract

Lactic acid bacteria are considered an inexhaustible source of bioactive compounds; indeed, products from their metabolism are known to have immunomodulatory and anti-inflammatory activity. Recently, we demonstrated that Cell-Free Supernatants (CFS) obtained from *Lactobacillus (*L*.) acidophilus*, *Lactiplantibacillus* (*L*.) *plantarum*, *Lacticaseibacillus* (*L*.) *rhamnosus,* and *Limosilactobacillus* (*L*.) *reuteri* can impair *Candida* pathogenic potential in an in vitro model of epithelial vaginal infection. This effect could be ascribed to a direct effect of living lactic acid bacteria on *Candida* virulence and to the production of metabolites that are able to impair fungal virulence. In the present work, stemming from these data, we deepened our knowledge of CFS from these four lactic acid bacteria by performing a metabolomic analysis to better characterize their composition. By using an untargeted metabolomic approach, we detected consistent differences in the metabolites produced by these four different lactic acid bacteria. Interestingly, *L. rhamnosus* and *L. acidophilus* showed the most peculiar metabolic profiles. Specifically, after a hierarchical clustering analysis, *L. rhamnosus* and *L. acidophilus* showed specific areas of significantly overexpressed metabolites that strongly differed from the same areas in other lactic acid bacteria. From the overexpressed compounds in these areas, inosine from *L. rhamnosus* returned with the best identification profile. This molecule has been described as having antioxidant, anti-inflammatory, anti-infective, and neuroprotective properties. The biological significance of its overproduction by *L. rhamnosus* might be important in its probiotic and/or postbiotic activity.

## 1. Introduction

Among probiotics, lactic acid bacteria are beneficial microbes for human health, when administered in adequate quantity [1,2]. As with all probiotics, lactic acid bacteria have effects on microbial pathogens and on the host. Specifically, lactic acid bacteria compete with pathogens for nutrients and binding to receptors and they also produce antimicrobial molecules. Their beneficial effects on the host include improvement in epithelial barrier function (through the enhanced production of mucus and of tight junction proteins that help to prevent the passage of the pathogens to the blood), the modulation of dendritic cell and T-cell activity (immunomodulatory effects), and the regulation of the production and secretion of several neurotransmitters [1,2]. In addition, lactic acid bacteria help to prevent and manage several pathological conditions, such as allergic diseases, cancer, hypercholesterolemia, irritable bowel syndrome, diarrhea, lactose intolerance, and inflammatory bowel disease [1,3]. One of the main roles played by probiotic lactic acid bacteria is to help the recovery of the eubiosis state in the host. However, the way this goal is achieved is partly unknown. In particular, the precise role of the metabolites produced by specific bacteria during their life cycle and their impact on the environment where they proliferate is yet to be elucidated. In addition, it must be considered that the use of living bacteria in vulnerable people is linked to possible safety concerns; also, maintaining bacterial viability is a challenging task [4]. Interestingly, new scientific evidence points out that the health benefits granted by lactic acid bacteria are not necessarily related to viable bacteria. Indeed, their metabolites or bacterial components, collectively indicated as postbiotics, may also be the driving force behind health promotion. Postbiotics have been shown to have several biological activities (antimicrobial, antioxidant, anti-inflammatory, anti-proliferative, and immunomodulatory). Moreover, numerous studies have suggested the significant potential of postbiotics for disease treatment [5]. The metabolites produced by lactic acid bacteria can inhibit the growth of pathogens [6,7]. In addition, during the interaction with the host and other microorganisms that dwell in the same host niche, the metabolites produced by “beneficial microbes” such as lactic acid bacteria may exert a significant impact to counteract the infection process [8,9]. Similarly to the living bacteria, the metabolites produced by probiotics have been demonstrated to have many beneficial effects on the host, such as improvement in barrier function, (stimulating the enhanced production of tight junctions’ proteins and mucous), the promotion of changes in the microbiota composition, and immunomodulatory and anti-inflammatory activities [8,10]. Since postbiotics are made up of inactivated microbial cells and/or cell components, their employment is characterized by higher levels of stability and safety for the user. Consequently, interest is increasing in their possible therapeutic employment, because they can be considered an inexhaustible source of possible new bioactive substances [11]. We recently showed that cell-free supernatants (CFS) obtained from *Lacticaseibacillus rhamnosus* (*L.* RHA), *Lactobacillus acidophilus* (*L.* AC), *Lactiplantibacillus plantarum* (*L.* PLA), and *Limosilactobacillus reuteri* (*L.* REU) can impair *Candida parapsilosis* (*C. parapsilosis*) pathogenic potential in an in vitro model of epithelial vaginal infection [12]. This effect could be ascribed to the direct effect of lactic acid bacteria on *Candida* virulence, and to the production of their metabolites that are able to weaken *C. parapsilosis* virulence [12]. Moreover, it has been recently shown that *L.* RHA can impair *C. albicans* pathogenicity in a model of intestinal epithelial infection. In this work, Alonso-Roman and coworkers showed that *L.* RHA growth alters the intestinal metabolic environment by removing *Candida* nutrient sources, forcing metabolic changes in *C. albicans* [9]. This suggests that the host niche colonization by specific bacteria can antagonize potential microbial pathogens by reshaping the metabolic environment and forcing microbial adaptation. Therefore, by improving our knowledge of the metabolome of beneficial microorganisms that can act within specific host niches, novel important information becomes available on the mechanisms they use to interact with the resident microbiota and with the host cells. For this reason, here, an untargeted metabolomics approach was applied to compare the metabolome of four different lactic acid bacteria often used as probiotics: *L.* RHA, *L.* AC, *L.* REU, and *L.* PLA. Our data show that such metabolomes are significantly different, resulting in an increased production of some specific metabolites, such as inosine, from *L.* RHA. Since inosine can exert antioxidant, anti-inflammatory, and neuroprotective effects [13], other than displaying relevant properties in the prokaryotic metabolisms, our data suggest that the overproduction of inosine by *L.* RHA could have a positive impact on the host and even on its resident microbiota. By employing an untargeted metabolomic approach, the present study shows that it is possible to predict the presence of compounds with potentially relevant biological activity, therefore accelerating knowledge regarding postbiotics. Indeed, although inanimate, postbiotics may provide health benefits comparable or even higher with respect to probiotics. Postbiotics include a wide range of microbial metabolites that could potentially produce complex beneficial effects by interacting with both resident microbiota and host cells. In addition, postbiotics could be considered as potentially novel therapeutics tools, even though evidence of the effect of postbiotics on microbiota and host cells is scant.

## 2. Materials and Methods

### 2.1. Lactic Acid Bacterial Strains and Growth Conditions

Four different lactic acid bacterial strains were employed in this study: *Lactobacillus acidophilus* ATCC 314, *Limosilactobacillus reuteri* DSM 17938, *Lacticaseibacillus rhamnosus* ATCC 7469, and *Lactiplantibacillus plantarum* ATCC 8014. The bacterial colonies were inoculated in 5 mL of MRS liquid medium (De Man, Rogosa and Sharpe, Oxoid LTD, Basingstoke, UK) and incubated for 24 h at 37 °C, under agitation. After incubation, bacteria were centrifuged at 2300× *g* at RT for 5 min, washed twice with PBS (Sial group), counted, and resuspended at 1 × 10^8^/mL in 5 mL of MRS broth and incubated for 24 h at 37 °C under agitation. After incubation, the cell-free supernatants (CFS) were prepared as detailed below.

### 2.2. Preparation of Cell-Free Supernatants (CFS) from Lactic Acid Bacterial Strains

The cell-free supernatants (CFS) were obtained by centrifugation of the bacterial suspensions carried out at 3000× *g*, at 4 °C for 15 min. The supernatants were then collected and filtered with 0.22 μm syringe filters (Corning Incorporated, Wiesbaden, Germany). Potential bacterial contamination of CFS was excluded by incubating 1 mL of each CFS at 37 °C and checking the turbidity (from 24 h to 72 h) by optical density (OD) assessment through spectrophotometer (SunRise, Tecan). The pH of each CFS was measured by a pH meter (Hanna Instrument, Villafranca Padovana, Italy), returning an average pH = 4, as previously described [12]. The control samples consisted of sterile MRS medium (blank). The CFS obtained were finally stored at −80 °C until their use.

### 2.3. Liquid Chromatography–Electrospray/High-Resolution Mass Spectrometry (HPLC-ESI/HRMS)

The CFS, which had been stored at −80 °C, were thawed and centrifuged at 18,000× *g* for 10 min. Subsequently, the CFS were transferred to Amicon-Ultra 0.5 tubes, centrifuged at 18,000× *g* for 15 min, and then transferred into the autosampler vials pending analysis. The Quality Control pool samples (QC) were prepared by mixing equal volumes of each cohort supernatant and used to minimize technical data variance [14].

The analyses were performed using an Ultimate 3000 HPLC connected to a QExactive High-Resolution Mass spectrometer via a HESI-II electrospray ionization source (Thermo Scientific, Waltham, MA, USA), controlled by Xcalibur software (Thermo Scientific, v. 29 build 2926). A 10 µL volume of sample solution was injected onto a Hypersil Gold C18 100 × 2.1 mm ID 1.9 µm ps column (Thermo Scientific) kept at 30 °C and separation was performed at 0.4 mL/min flow with a gradient elution scheme using methanol (Fisher Chemicals, Hampton, NH, USA) (B) and 0.1% formic acid (Carlo Erba, Cornaredo, Italy) in water (A). The mobile phase composition was kept at 2% B for 1 min after injection then linearly raised to 42% B in 60 min and further on to 98% B in 5 min. Methanol was kept at 98% up to minute 74.9, then lowered to 2% at minute 75. The total runtime was 90 min. ESI source was operated in both positive and negative ionization mode. Capillary temperature was set at 320 °C; the following nitrogen flows (arbitrary units) were used to assist the ionization: Sheath Gas 45, Aux Gas 25 (at 290 °C), Sweep Gas 2. The capillary voltage was set to 3.8 kV (3.4 kV for negative ionization) and S-Lens RF level was set at 45 (arbitrary units).

A Data-Dependent Acquisition (DDA) strategy was used to acquire MS2 fragmentation spectra of the Top 5 singly charged precursor ions revealed in Full Scan MS experiments. Positive and Negative ionization DDA experiments were performed in separate analyses. Full MS spectra were obtained from *m*/*z* 100 to 1500 at 70,000 FWHM resolving power using an automatic gain control (AGC) of 3 × 106 and a maximum Injection Time (max IT) of 250 ms. Fragmentation Spectra (MS2) acquisition was performed at 17,500 FWHM, with 2 × 105 AGC target and 120 ms max IT. The isolation window for precursor ion selection was set at 1.0 Th and HCD normalized collision energy (NCE) was stepped at 20, 50, and 80. Fragmented precursors were dynamically excluded for 6 s. Inosine standard was purchased from Sigma-Aldrich, St. Louis, MO, USA. The data are from triplicate samples from 3 different experiments.

### 2.4. Compounds Discoverer Data Analysis

Raw files (triplicate samples from 3 different experiments) were processed by Compound Discoverer (CD) 3.3.2.31 (Copyright 2014-2023 Thermo Fisher Scientific Inc.) using a slightly modified processing workflow template for Untargeted Metabolomics with Statistics Detect Unknowns with ID Using Local Databases. The core of the workflow consisted of Spectra selection from raw files (Retention Time limited from 0.2 to 75 min), Retention Time Alignment (ChromAlign) with respect to a QC sample file, and Compound Detection and Grouping with RT tolerance of 0.3 min and 5 ppm mass deviation. Then, Gap Filling, SERRF QC Correction, and Background removal were performed along with Compound Annotation using Predicted Composition and different types of databases (mzCloud, Metabolika, Human Metabolome Database, ChemSpider, BioCyc) [15]. The so-detected compounds were used for differential analysis of sample groups (Nested Design; Generated Ratios: lactobacilli PLA/AC, REU/AC, RHA/AC, REU/PLA, RHA/PLA, and RHA/REU).

## 3. Results

Here, an untargeted metabolomics approach was used to compare the metabolomes from four different lactic acid bacteria, often used as probiotics: *L. rhamnosus* (*L.* RHA), *L. acidophilus* (*L.* AC), *L. plantarum* (*L.* PLA), and *L. reuteri* (*L.* REU). Principal Component Analysis (PCA)-2 showed that the metabolomes differed between *L.* RHA and the other species, as well as between *L.* AC and the other lactic acid bacteria. Conversely, the metabolomes of *L.* PLA and *L.* REU were found to be similar (Figure 1).

A hierarchical clustering analysis, carried out to compare the four metabolomes, revealed a distinct cluster of metabolites overexpressed in the CFS of the different lactic acid bacteria. Once again, *L.* PLA and *L.* REU showed similar metabolome profiles; differently, *L.* RHA and *L.* AC showed a more peculiar metabolome profile (Figure 2). Specifically, *L.* RHA and *L.* AC revealed areas of significantly overexpressed metabolites (*p* value < 0.01; Log2 fold change = 2) that strongly differed from the same areas from the other lactic acid bacteria (Figure 2, see red line for *L.* RHA and yellow line for *L.* AC).

Therefore, we performed a more detailed analysis of the metabolites included in these specific areas of *L.* RHA and *L.* AC. The identified overexpressed compounds in these areas returned different levels of identification according to the Annotation Sources used by the CD software 3.3.2.31 and listed in Table 1, Table 2, Table 3 and Table 4. Only those compounds that the software assigned a name to are reported in Table 1, Table 2, Table 3 and Table 4.

According to the results of the analysis, inosine from *L.* RHA CFS returned the best identification profile, since it returned four Full Matched and two Partial Matches according to the Annotation Sources as shown in Table 1. Concerning *L.* AC, we found two compounds with a very good identification profile (three Full Matches and one Partial Match): Acetylcholine and Indole-3-lactic acid.

Table 5 shows the identified pathways that included inosine. For each pathway, the mapped and matched compounds and the total compounds in the pathway are shown.

Interestingly, this molecule has also a well-known biological role. Indeed, inosine is a key intracellular energy substrate for nucleotide synthesis by salvage pathways and it possesses cell protective activity and cell repair properties [16].

To increase the identification confidence over inosine, from probable to possibly confirmed structure [17], an inosine reference standard solution was used to confirm the [M+H]^+^ molecular ion mass-to-charge ratio, along with its fragmentation spectrum and retention time (Figure 3 and Figure 4).

Once the identification was confirmed, inosine from *L.* RHA CFS was quantified using a set of calibration samples obtained by adding a proper amount of inosine to MRS covering from 1 to 50 µg/mL concentration range. Inosine of the *L.* RHA sample was quantified in the range of 5–8 µg/mL.

## 4. Discussion

Lactic acid bacteria are beneficial microbes, and they are often used as probiotics. The concept of probiotics has been evolving, with the currently accepted definition being “living microorganisms that can benefit the host when consumed in sufficient quantities” [18]. Although this definition implies that microorganisms must be viable to be beneficial, increasing evidence suggests that microbial products can also provide benefits to the host [19,20]. Indeed, the so-called postbiotics, also known as metabolites, biogenic or cell-free supernatants (CFS), are defined as “soluble factors secreted by living bacteria or released by bacterial lysis”, and their role in providing health benefits to the host has been reported [21]. Hence, today, it is acknowledged that the beneficial effects of lactic acid bacteria are based either on living bacteria (the “probiotics”) or on their metabolites/cell lysates (the “postbiotics”). Here, we assess the metabolomic profiles of four different lactic acid bacteria, currently used as safe probiotics: *L.* RHA, *L.* AC, *L.* PLA, and *L.* REU [22]. An untargeted metabolomic approach was employed to compare the differences in metabolite production in the CFS from the different lactic acid bacteria under the same culture conditions. Specifically, hierarchical clustering analysis of the compounds released by the four lactic acid bacteria shows for *L.* RHA and *L.* AC specific areas of significantly overexpressed metabolites, which strongly differ from the same areas of the other lactic acid bacteria. It has been shown that CFS from *L.* RHA strain SCB0119 altered the transcription profiles of several genes involved in fatty acid degradation, ion transport, and the biosynthesis of amino acids in *Escherichia coli*, as well as fatty acid degradation, protein synthesis, DNA replication, and ATP hydrolysis in *Staphylococcus aureus*, which are important for bacterial survival and growth [23]. In addition, *L.* RHA colonization of the epithelial cells has been demonstrated as being responsible for drastic changes in the metabolic environment, forcing metabolic adaptation in *C. albicans* and reducing fungal virulence [9]. Furthermore, antimicrobial properties of *L.* RHA have also been described against *Listeria monocytogenes* [24] and *Salmonella* spp. [25,26]. *L.* AC has been found to play important roles in many aspects of human health. It favors the eubiosis of the host intestinal tract through the production of metabolites. Among the molecules produced by *L.* AC, lactic acid is important to reduce the pH, which, in turn, inhibits the growth and virulence of pathogenic bacteria [27].

Therefore, according to all the above-mentioned literature data, reporting the antimicrobial activity of *L.* RHA and *L.* AC [27,28,29,30], a detailed investigation was carried out to obtain more information on metabolite overexpressed by these species. Among the overexpressed compounds, we identified inosine from the CFS of *L.* RHA as the molecule with the best identification profile. We also identified four molecular pathways, including inosine, that could be investigated in future studies. Inosine is a non-canonical nucleotide, mainly occurring in the form of a monophosphate. It base pairs with deoxythymidine, deoxyadenosine, and deoxyguanosine [31]. Among the possible roles of such an unconventional nucleotide, it has been reported that the incorporation of inosine in place of guanine modulates translational events [32]. Several studies carried out in various neuronal cell types have identified the growth-promoting activity of inosine, comparable to that induced by canonical neurotrophic factors such as brain-derived neurotrophic factor (BDNF) or nerve growth factor (NGF) [33,34]. Benowitz and colleagues have shown that inosine promotes axon outgrowth in a rat model of corticospinal tract injury [35]. Furthermore, inosine has been demonstrated to modulate several biological processes through the adenosine receptors, such as the enhancement of neurite outgrowth in depressive disorders [36]. Because of its antioxidant, anti-inflammatory, pro-axogenic, and neuroprotective functions, inosine is also employed as a therapeutic supplement, and it is prescribed in cases of nerve injury, inflammation, and oxidative stress [13,37]. In addition, several drugs used in the treatment of autoimmune and inflammatory diseases (such as adenosine kinase inhibitors) exert their beneficial effects by releasing adenosine [38]. Since the latter is readily degraded to inosine in the extracellular space, the direct involvement of inosine in the anti-inflammatory effects of these adenosine-releasing agents is conceivable [38]. Inosine has also immunomodulatory effects by contributing to the efficacy of Isoprinosine (inosine pranobex), a synthetic agent formed by inosine combined with the immunostimulant dimepranol acedoben (acetamidobenzoic acid and dimethylaminoisopropanol). Even though many of the biological actions of inosine (particularly in the context of microbial infections) have yet to be described, this molecule is already employed for the treatment of acute respiratory viral infections, genital warts, herpes simplex infections, hepatitis B, and subacute sclerosing panencephalitis [13]. Inosine is used also for the treatment of sepsis in infections, and it has been shown to reduce systemic inflammation, organ damage, tissue dysoxia, and vascular dysfunction, resulting in improved survival in a mouse model of septic shock [39]. Therefore, according to our preliminary in vitro data demonstrating that *L.* RHA produces biologically significant amounts of inosine, future studies will be devoted to assessing if inosine production also occurs in vivo. In addition, it will be necessary to confirm its anti-inflammatory, antioxidant, and antimicrobial activities in ex vivo and in vivo infection models. The overproduction of inosine by *L.* RHA in an in vivo setting might explain the beneficial effect of *L.* RHA as a probiotic because it might set off a complex intertwining network (where inosine could be one of the key players within the several pathways identified) with beneficial effects on both resident microbiota and host cells.

## 5. Conclusions

In conclusion, here, we show that *L.* RHA overproduces inosine during its life cycle, and this might have a significant impact when administered in vivo. The data shown in the present manuscript were generated by an in vitro experimental system, supplemented by an extremely thorough in silico metabolomic analysis. The limitation of the present study is that the analysis was carried out with only one *L.* RHA strain. For this reason, future studies are warranted to confirm if such an overproduction of inosine can also be observed in other *L.* RHA strains. In addition, it will be important to translate these very interesting preliminary results in ex-vivo systems. Furthermore, it will be important to assess and contextualize the effects of inosine produced by *L.* RHA in infection models. Finally, our experimental approach should be applied to the study of the metabolome of other lactic acid bacteria to identify metabolites involved in their postbiotic activities.

## Figures and Tables

**Figure 1 microorganisms-12-00662-f001:**
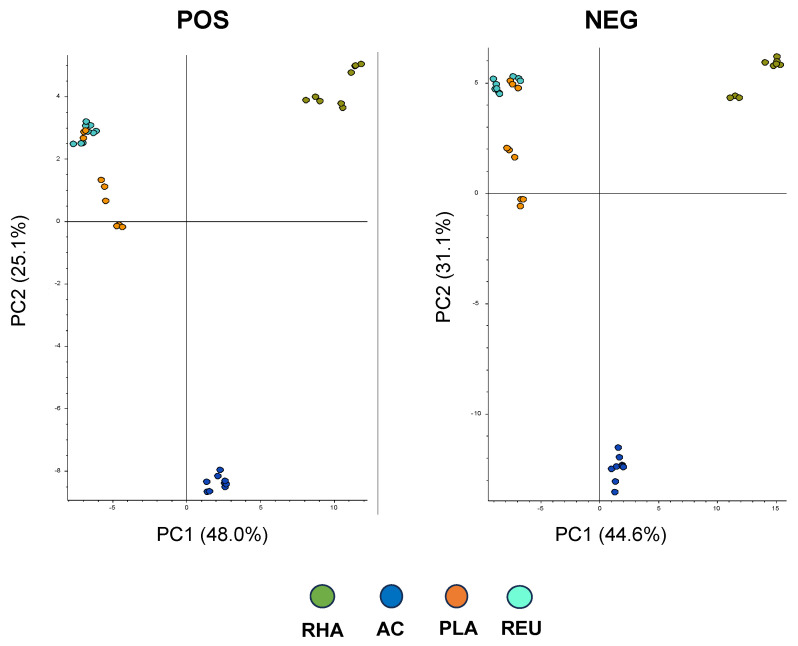
Principal Component Analysis (PCA)-2 of the metabolomes from *L.* RHA, *L.* AC, *L.* PLA, and *L.* REU analyzing in both positive and negative ionization modes. Data are from triplicate samples from 3 different experiments.

**Figure 2 microorganisms-12-00662-f002:**
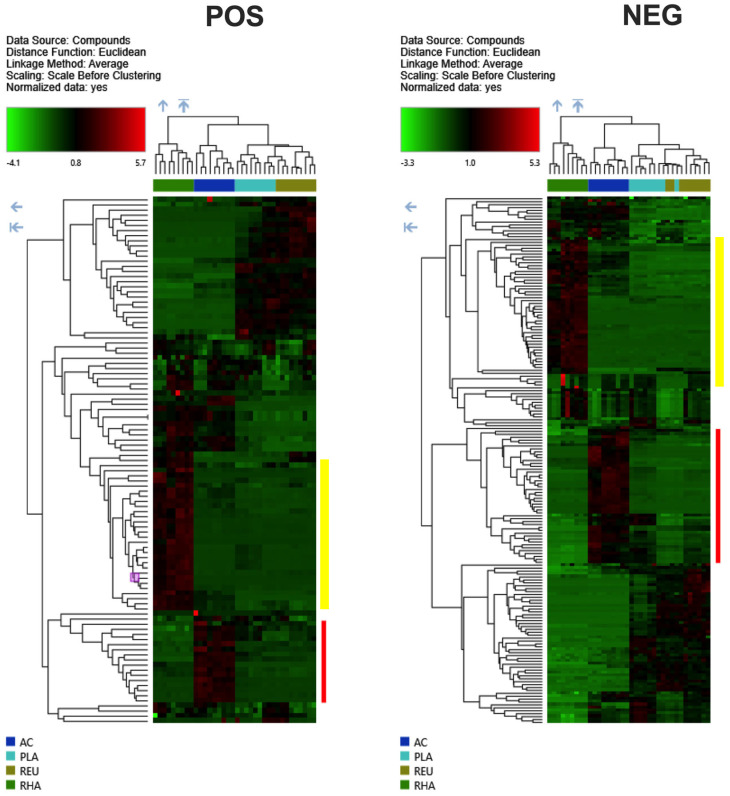
Hierarchical clustering analysis in positive (**left panel**) and negative (**right panel**) ionization mode, carried out to compare the metabolome from *L.* RHA, *L.* AC, *L.* PLA, and *L.* REU according to Compound Discoverer (CD) 3.3.2.31 analysis. Yellow line for *L.* RHA and red line for *L.* AC highlight the distinct cluster of metabolites overexpressed in the respective CFS as compared to the other lactic acid bacteria CFS. Data are from triplicate samples from 3 different experiments.

**Figure 3 microorganisms-12-00662-f003:**
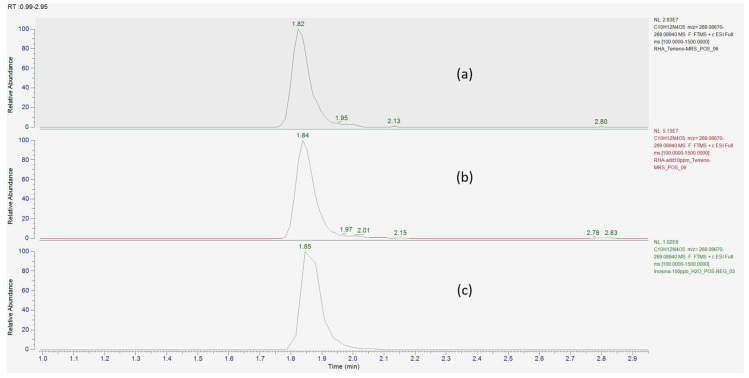
Extracted ion chromatogram of inosine (C_10_H_12_N_4_O_5_) theoretical [M+H]^+^ molecular ion at *m*/*z* = 269.08805 (±5 ppm) in (**a**) *L.* RHA CFS, (**b**) *L.* RHA CFS spiked with inosine, and (**c**) inosine standard solution.

**Figure 4 microorganisms-12-00662-f004:**
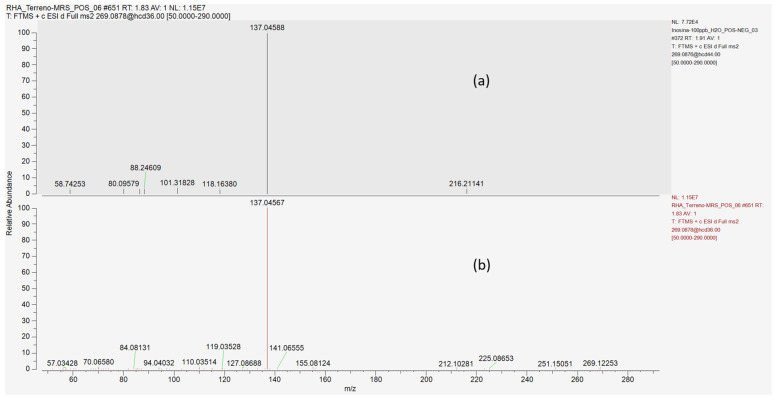
Comparison of HCD fragmentation spectra of *m*/*z* = 269.08805 precursor ion at 1.83 min in (**a**) inosine standard solution and (**b**) *L.* RHA CFS.

**Table 1 microorganisms-12-00662-t001:** *L.* RHA overexpressed compounds identified in positive ionization mode according to Compound Discoverer (CD) 3.3.2.31 analysis. The table includes only those molecules to which the software could assign a name. The dash indicates no results or invalid mass. Data are from triplicate samples from 3 different experiments.

Compound Name	Predicted Composition	mzCloud Search	mzValue Search	Metabolika Search	ChemSpider Search	MassList Search
Mevalonolactone	Full Match	Full Match	-	Partial Match	Full Match	Partial Match
N-α-L-Acetyl-arginine	Full Match	Full Match	-	-	Full Match	Full Match
(Ac)2-L-Lys-D-Ala	Full Match	-	-	-	Full Match	-
**Inosine**	**Full Match**	**Full Match**	**Full Match**	**Full Match**	**Partial Match**	**Partial Match**
(2S,4S)-4-Amino-2-hydroxy-2-methylpentanedioic acid	Full Match	-	-	-	Full Match	-
Confluenine A	Full Match	-	-	-	Partial Match	Full Match
Ala-Leu	Full Match	-	-	-	Full Match	-
O-Succinyl-L-homoserine	Full Match	-	-	Partial Match	Partial Match	Full Match
N-Acetyl-L-leucine	Full Match	Full Match	-	-	Full Match	Full Match
(+)-Flavipucine	Full Match	-	-	-	Partial Match	Full Match
2-(2-amino-3-methylbutanamido)-3-phenylpropanoic acid	Full Match	Full Match	-	-	Partial Match	Partial Match
Acetyl-L-carnitine	Full Match	-	-	-	Partial Match	Full Match
methyl 4-(3,4-dihydroxybenzamido)butanoate	Full Match	-	-	-	-	Full Match
3-amino-4,5,6-trihydroxy-2-methoxy-5-methyl-2-cyclohexen-1-one	Full Match	-	-	-	Partial Match	Full Match
(2S)-2-{[1-(R)-Carboxyethyl]amino}pentanoate	Full Match	-	-	-	Partial Match	Full Match
2-Amino-5-methyl-5-hexencarbonsaeure	Full Match	-	-	-	Partial Match	Full Match
(1S,2R,8aS)-1,2-Dihydroxyindolizidine	Full Match	-	-	-	Full Match	Full Match

The color code mirrors the color code of the software.

**Table 2 microorganisms-12-00662-t002:** *L.* AC overexpressed compounds identified in positive ionization mode according to Compound Discoverer (CD) 3.3.2.31 analysis. The table includes only those molecules to which the software could assign a name. The dash indicates no results or invalid mass. Data are from triplicate samples from 3 different experiments.

Compound Name	Predicted Composition	mzCloud Search	mzValue Search	Metabolika Search	ChemSpider Search	MassList Search
Acetylcholine	Full Match	Full Match	-	-	Partial Match	Full Match
N-Acetyl-S-2-hydroxyethyl-L-cysteine	Full Match	-	-	-	Full Match	-
Palmyrrolinone	Full Match	-	-	-	Partial Match	Full Match
Indole-3-lactic acid	-	Full Match	-	Partial Match	Full Match	Full Match
Gaburedin A	Full Match	-	-	-	Partial Match	Full Match
3-amino-4,5,6-trihydroxy-2-methoxy-5-methyl-2-cyclohexen-1-one	Full Match	-	-	-	Partial Match	Full Match
Louisianin B	Full Match	-	-	Partial Match	Partial Match	Full Match
γ-L-glutamyl-L-leucine	-	-	-	-	-	Full Match
Sistodiolynne	Full Match	-	-	Partial Match	Partial Match	Full Match
Quinoline-2-methanol	Full Match	-	-	-	Partial Match	Full Match

**Table 3 microorganisms-12-00662-t003:** *L.* RHA overexpressed compounds identified in negative ionization mode according to Compound Discoverer (CD) 3.3.2.31 analysis. The table includes only those molecules to which the software could assign a name. The dash indicates no results or invalid mass. Data are from triplicate samples from 3 different experiments.

Compound Name	Predicted Composition	mzCloud Search	mzValue Search	Metabolika Search	ChemSpider Search	MassList Search
Mevalonic acid	-	-	-	-	-	-
O-Succinyl-L-homoserine	Full Match	-	-	Partial Match	Partial Match	Full Match
N-Acetyl-D-alloisoleucine	-	-	-	-	-	-
O-Succinyl-L-homoserine	Full Match	-	-	Partial Match	Partial Match	Full Match
N-Acetylvaline	-	-	-	-	-	-
Birnbaumin A	Full Match	-	-	-	-	Full Match
methyl 4-(3,4-dihydroxybenzamido)butanoate	Full Match	-	-	-	-	Full Match
3,11-dihydroxy-6,8-dimethyldodecanoic acid	Full Match	-	-	-	-	Full Match
Ala-Leu	-	-	-	-	Full Match	-
Taurochenodeoxycholic acid	-	-	-	-	-	-
Phenamide	Full Match	-	-	-	Partial Match	Full Match
Lorbamate	Full Match	-	-	-	Full Match	-
Pochonicine	Full Match	-	-	-	-	Full Match

The color code mirrors the color code of the software.

**Table 4 microorganisms-12-00662-t004:** *L.* AC overexpressed compounds identified in negative ionization mode according to Compound Discoverer (CD) 3.3.2.31 analysis. The table includes only those molecules to which the software could assign a name. The dash indicates no results or invalid mass. Data are from triplicate samples from 3 different experiments.

Compound Name	Predicted Composition	mzCloud Search	mzValue Search	Metabolika Search	ChemSpider Search	MassList Search
3-Phenyllactic acid	-	-	-	-	-	-
DL-4-Hydroxyphenyllactic acid	-	Full Match	Full Match	Full Match	-	-
2-Hydroxycaproic acid	-	-	-	-	-	-
2-Hydroxyvaleric acid	-	-	-	-	-	-
trans-Cinnamic acid	-	-	-	-	-	-
(+)-Flavipucine	Full Match	-	-	-	Partial Match	Full Match
Indole-3-lactic acid	-	Full Match	-	Partial Match	-	-
zidometacin	Full Match	-	-	-	Full Match	-
(+)-(17R)-apralactone A	Full Match	-	-	-	-	Full Match
2-Oxoglutaric acid	-	-	-	-	-	-
N-Acetyl-L-glutamine	-	Full Match	Full Match	-	-	-
2-Hydroxycinnamic acid	-	-	-	-	-	-
Pochonicine	Full Match	-	-	-	-	Full Match
Peniamidone B	Full Match	-	-	-	-	Full Match
2-Methoxyestradiol	-	-	-	-	-	-
Hopantenic acid	Full Match	-	-	-	Full Match	-
Pantetheine	Full Match	-	-	-	Partial Match	-
methyl 4-(3,4-dihydroxybenzamido)butanoate	Full Match	-	-	-	-	Full Match

The color code mirrors the color code of the software.

**Table 5 microorganisms-12-00662-t005:** Inosine identified pathways according to Compound Discoverer (CD) 3.3.2.31 analysis. Data are from triplicate samples from 3 different experiments.

Compound Name	Formula	n° Identified Pathways	Pathways	Mapped Compounds	Matched Compounds	Compoundsin Pathways
Inosine	C_10_H_12_N_4_O_5_	1	Superpathway of purine nucleotide salvage	14	10	54
		2	Purine nucleotides degradation II (aerobic)	12	8	27
		3	Purine nucleotides degradation I (plants)	10	7	23
		4	Superpathway of purine degradation in plants	10	7	34

## Data Availability

Data are contained within the article.

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
