# Peer review of "An Untargeted Metabolomic Analysis of Lacticaseibacillus (L.) rhamnosus, Lactobacillus (L.) acidophilus, Lactiplantibacillus (L.) plantarum and Limosilactobacillus (L.) reuteri Reveals an Upregulated Production of Inosine from L. rhamnosus"

_microorganisms, 2024, doi:10.3390/microorganisms12040662_

Round 1

Reviewer 1 Report

Comments and Suggestions for Authors

The article presents interesting and scientifically valuable data on the metabolomes of four commonly used types of lactic acid. The study has some limitations. For example, the study focuses on metabolites and does not examine the influence of inosine on the probiotic effect of L. rhamnosus in vivo. However, authors should consider clarifying certain information to reinforce the significance of their findings.

·         In general – Please use the current names of lactic acid bacteria in the manuscript: Lactiplantibacillus plantarum, Lacticaseibacillus rhamnosus, and Limosilactobacillus reuteri; see: lactobacillus.ualberta.ca.

·         1. Introduction – The introduction begins with a comprehensive overview of probiotics and then narrows down to a specific focus on metabolites and postbiotics. Consider removing some unnecessary information. In the last sentence of the introduction, clearly state the research question or hypothesis of your study. You can explicitly state the research question you are addressing in this study. Focus on the novelty of your work. Please briefly mention the limitations of existing research on postbiotics and how your study addresses these.

·         2.1. Lactobacillus strains and growth conditions – Please describe how anaerobic conditions were ensured during the stirring incubation.

·         2.2. Preparation of cell-free supernatants (CFS) from Lactobacillus strains – Please calculate " rpm " as "x g".

·         2.3. Liquid Chromatography – Electrospray/High Resolution Mass Spectrometry (HPLC-116 ESI/HRMS) – Please calculate " rpm " as "x g".

·         3. Results – Please briefly explain the purpose of each analysis and its key findings. Please focus on the key findings and avoid unnecessary repetition. Some information could be compressed. For example, the detailed explanation of the annotation sources could be omitted. The explanation of principal component analysis (PCA) and hierarchical cluster analysis could be simplified. Consider removing some unnecessary information, such as: B. repeated mentions of positive and negative ionization modes. Please briefly mention any other important results, but focus primarily on inosine. The section could focus more on the specific findings related to inosine. The section does not mention the results for L. acidophilus. Were there other significantly overexpressed metabolites in this strain?

·         4. Discussion – Please briefly summarize the key findings of the study before discussing its implications. Remove unnecessary information and redundancy. Improve the flow of ideas by connecting different paragraphs more smoothly. Discuss the possible mechanisms by which inosine produced by L. rhamnosus might exert its beneficial effects. Are there any known pathways or interactions that could be examined in future studies? Emphasize the novelty and potential impact of your findings. How does your study contribute to the understanding of L. rhamnosus and its potential health benefits? I think you could mention possible limitations of your study, such as the use of in vitro models or the lack of direct evidence for the role of inosine in probiotic effects. Please briefly discuss future research directions to investigate the potential benefits of inosine and its role in L. rhamnosus function.

·         5. Conclusions – Please mention the specific potential benefits of inosine overproduction in L. rhamnosus based on its known properties. Consider how the finding of inosine overproduction relates to the broader goal of understanding the lactobacilli metabolome. Be more specific about the potential impacts and why further research is important. Emphasize the novelty and importance of your findings.

·         Abstract – In my opinion, there is no need to mention the specific anti-Candida effect as it is not directly relevant to the focus of the current study (inosine). Consider simplifying the first sentence by removing unnecessary language such as "potentially bioactive substances" and "several". Simply state that lactobacilli produce metabolites with beneficial effects. Instead of listing them all, briefly mention the most important properties of inosine. Instead of mentioning all four lactobacilli, highlight the key finding of inosine overproduction by L. rhamnosus. Highlight the specific contribution of your study to the understanding of L. rhamnosus and its potential benefits.

·         Keywords – Please consider adding a keyword related to "postbiotics" as this is an important aspect of the study.

Reviewer 2 Report

Comments and Suggestions for Authors

This is a quite sophisticated study applying metabolomic analysis of a few Lactobacillus strains focusing on the production of inosine. Introduction justifies the study, methods which were applied are appropriate, results are clearly represented and discussed. However, introduction should end up with aiming the target of your study, asking questions which you want to answer and not presenting the results! So, you should definitely change the end of your introduction part.

I also have some doubts about your conclusions and what you suggest in the title of your paper. You only analyzed one strain of L. rhamnosus. Do you think that's all strains of L. rhamnosus will overproduce inosine? Probably, but we cannot be so sure. In the perfect world you should study more strains of L. rhamnosus but I know how it is when you finish a certain stage of your study, it's really hard to come back and do additional analysis. So, you should discuss it at least, you can also indicate weakness and strength of your study or change the title or conclusions at least. Then your paper will be honest and can be published.

Minor remarks below.

Numbers up to 10 not followed by a unit should be written in words (e.g. line 24)

Apply italics to Latin expressions, e.g. in silico line 303

Reviewer 3 Report

Comments and Suggestions for Authors

Overall, the manuscript presents a comprehensive investigation into the metabolomic profiles of four different lactobacilli species, focusing on their potential antimicrobial activities and the identification of key metabolites, such as inosine. It is presented in a well-structured manner and makes a valuable contribution to the field of probiotics and microbial metabolomics.

Please mind the comments below:

Keywords

Try to use keywords that are not reported in the manuscript title to increase search possibilities.

Introduction

Line 83: "suggest" should be corrected to "suggests".

Materials and Methods

Line 103: Please specify the anaerobic conditions. Was an anaerobic jar with gas packs used or an anaerobic chamber?

Line 103: Please specify centrifugation conditions.

Line 103: Please provide information about the PBS supplier.

Line 109: Please use g instead of rpm throughout the manuscript.

Line 112: Please specify the equipment used for turbidity checking.

Line 129: Please provide information about methanol and formic acid supplier.

Please specify the software utilized for Principal Component Analysis and for statistical calculations, such as p-values.

Results

Line 172: It would be beneficial to provide the complete species of the microorganisms. This should be also applied to all Figures and Tables.

Discussion

While Table 5 in the Results section provides valuable insight into pathways involving inosine, further elaboration on the clinical relevance of these pathways would strengthen the discussion.

Conclusions

The conclusions could be strengthened by emphasizing the biological significance of inosine, in order to underscore potential avenues for future research aimed at elucidating the mechanistic underpinnings of inosine-mediated effects.

Comments on the Quality of English Language

The text should be checked for grammatical and syntax errors.

Round 2

Reviewer 1 Report

Comments and Suggestions for Authors

The author responded to all the queries. I do not have any questions.

Reviewer 2 Report

Comments and Suggestions for Authors

I accept the changes